# The Interplay between Endocrine-Disrupting Chemicals and the Epigenome towards Metabolic Dysfunction-Associated Steatotic Liver Disease: A Comprehensive Review

**DOI:** 10.3390/nu16081124

**Published:** 2024-04-11

**Authors:** Evangelia Mentsiou Nikolaou, Ioanna Panagiota Kalafati, George V. Dedoussis

**Affiliations:** 1Department of Nutrition and Dietetics, School of Health and Education, Harokopio University of Athens, 17676 Athens, Greece; evemen@hua.gr (E.M.N.); dedousi@hua.gr (G.V.D.); 2Department of Nutrition and Dietetics, School of Physical Education, Sport Science and Dietetics, University of Thessaly, 42132 Trikala, Greece

**Keywords:** metabolic dysfunction-associated steatotic liver disease (MASLD), endocrine-disrupting chemicals (EDCs), diet, epigenetic changes, DNA methylation, histone alterations, microRNAs, transgenerational inheritance, gene–environment interactions

## Abstract

Metabolic dysfunction-associated steatotic liver disease (MASLD), described as the most prominent cause of chronic liver disease worldwide, has emerged as a significant public health issue, posing a considerable challenge for most countries. Endocrine-disrupting chemicals (EDCs), commonly found in daily use items and foods, are able to interfere with nuclear receptors (NRs) and disturb hormonal signaling and mitochondrial function, leading, among other metabolic disorders, to MASLD. EDCs have also been proposed to cause transgenerationally inherited alterations leading to increased disease susceptibility. In this review, we are focusing on the most prominent linking pathways between EDCs and MASLD, their role in the induction of epigenetic transgenerational inheritance of the disease as well as up-to-date practices aimed at reducing their impact.

## 1. Introduction

Metabolic dysfunction-associated steatotic liver disease (MASLD) is the new approved nomenclature to replace the non-alcoholic fatty liver disease (NAFLD) term, in order to broaden the diagnostic criteria and avoid stigmatisation of the patients [1]. MASLD, which is strongly associated with metabolic disorders such as obesity, type 2 diabetes mellitus (T2DM), and insulin resistance (IR), currently affects 2 out of 5 people worldwide, and its prevalence has grown by 50% over the last two decades. Furthermore, it has been reported as the first cause of liver injury, cancer, and transplantation. Molecular mechanisms in MASLD mainly encompass alterations in lipid metabolism, including de-novo lipogenesis upregulation, reduced β-oxidation and very-low-density lipoprotein (VLDL) secretion, mitochondrial dysfunction in the liver, and epigenetic changes including microRNA (miRNA) dysregulation [2]. MASLD management remains an intricate process requiring the involvement of multidisciplinary approaches.

Over the last four decades, endocrine-disrupting chemicals (EDCs) have been mass produced, and humans have encountered them through the food supply, everyday products, plastic containers, and industrial chemicals [3,4]. Some EDCs have been widely recognised as obesogens, contributing to obesity as well as other metabolic diseases in humans [4]. Moreover, many EDCs are metabolised in the liver; they appear to be closely associated with MASLD predisposition and also to cause significant epigenetic alterations that are inherited across generations, including DNA methylation, histone modification, and miRNA regulation [5,6]. These alterations lead to modified gene transcription/silencing, resulting in heritable changes in the epigenome that evidently persist across generations [4]. Moreover, EDCs interfere with hormonal signaling by disrupting the normal pathways, resulting in inflammation, lipid accumulation, and reactive oxygen species (ROS) overproduction, thus leading to abnormal function and disturbed homeostasis in the liver. Furthermore, numerous studies have reported that some EDCs might target mitochondria, leading to increased vulnerability of their DNA [7]. Herein, this is the first comprehensive review to summarise the literature bridging the topics of MASLD onset with exposure to EDCs, the related molecular mechanisms, and the subsequent epigenetic alterations, as well as current daily recommendations to reduce the impact of EDCs. This could pave the way to producing new non-invasive treatment options for MASLD by focusing on lifestyle factors and the alterations produced in the epigenome.

## 2. Materials and Methods

The relevant information concerning dietary treatment, EDC exposure, molecular pathways, and epigenetic alterations in MASLD was obtained from literature research through publicly available databases such as PubMed, Scopus, and Google Scholar. For the research, key terms were used, including “non-alcoholic fatty liver disease”, “Metabolic Dysfunction-Associated Steatotic Liver Disease”, “endocrine disrupting chemicals”, “microRNA”, “DNA methylation”, “Histone modifications”, ‘’epigenetics” and “diet/dietary approaches”. The most pertinent data have been considered, and the findings are outlined in the subsequent sections. Mainly, the focus was shifted to articles written in English and published within the past ten years, although without using a specific publication year cutoff.

## 3. Definitions and Types of EDCs

Studies have shown that EDCs, when introduced into the human body, can interfere with the endocrine system and lead to metabolic and immune dysfunctions, contributing to a wide range of non-communicable diseases, as well as developmental, reproductive, and neurological abnormalities [8]. The definition of an EDC by the World Health Organization is “an exogenous substance or mixture that alters the function(s) of the endocrine system and consequently causes adverse effects in an intact organism, or its progeny or (sub) population’’ [9]. EDCs are either naturally produced by plants or fungi, or are human made, mainly found in industrial procedures, pollutants, or everyday products including plastic bottles, cosmetics, food cans, pesticides, and cleaning agents [3]. Usually, exposure takes place through water consumption, by inhaling polluted air, the skin, the eyes, food ingestion, or by coming into contact with polluted soils [8].

It is estimated that nearly 1000 of today’s known chemicals meet the criteria of an EDC [10]. The five most extensively studied classes of EDCs will be briefly described below [11]. Firstly, the category of phytoestrogens, such as genistein and resveratrol, mainly found in bread, cereals, soy, legumes, and fruits. They are widely known for their antioxidant properties and their role in mimicking or antagonising the effects of oestrogens in the body [11,12]. Secondly, the category of industrial chemicals such as pesticides, flame retardants, and other products that include dioxin and dichlorodiphenyltrichloroethane (DDT). Another large source of EDCs derives from household and consumable products such as food and beverage packaging material, bottles, cosmetics, and other formulas. These products are rich in phthalates, bisphenol A (BPA), bisphenol F (BPF), and bisphenol S (BPS) [13,14]. Furthermore, the category of medical devices consists of disposable gloves, plastic devices, and intravenous tubing, which also contain BPA and phthalates. Lastly, it has been reported that the use of estrogenic pharmaceuticals can spread natural or synthetic steroids or diethylstilbestrol (DES), leading to adverse health effects related to fertility and development [14].

It is believed that around 800 commercial substances interfere with the immune system; thus, only a few of them have been recognised for their hazardous effects on human health [15]. Human exposure to EDCs, especially at extremely low doses, is widespread and involves various combinations leading to unpredictable potential effects. These effects may not be adequately addressed when assessing individual compounds per se. Hence, evaluating the potential risks to human health, arising from both individual and mixture exposures to EDCs, is a crucial and primary concern for ensuring safety.

## 4. Relation between Dietary Habits, EDCs and MASLD

Diet is one of the greatest sources of daily human exposure to EDCs [15]. Precisely, EDCs can infiltrate the food supply through various means, including food contact materials (such as plastic tubing, packaging materials) and environmental sources (such as animal feed, fertilizers, and polluted groundwater) [16]. In fact, most of these substances are not mentioned on food labels, thus making it challenging for individuals to control their exposure.

Food products can be tainted with EDCs, which normally exist in the packaging, the gloves used in food preparation, and the plasticised polyvinyl chloride (PVC) materials often used as food containers [17,18,19]. Higher EDC levels have been documented in animal products, such as poultry, and dairy items such as cream and butter [20]. This is attributed to the lipophilic nature of some EDCs, such as bis (2-ethylhexyl) phthalate (DEHP), which allows them to accumulate in fat-rich foods. Seafood products, particularly frozen or packaged, constitute another significant source of phthalates and BPA [18]. Intriguingly, fresh products such as fruits, vegetables, and pasta, as well as milk, yogurt, and eggs, are generally associated with lower phthalates and BPA levels, whereas canned or packaged fruits and vegetables tend to contain higher levels [18]. In fact, Koch H.M. et al., in their human biomonitoring study, found that exposure to high-molecular-weight phthalates appears to be driven by dietary intake, whereas exposure to low-molecular-weight phthalates is mainly driven by personal care products and other sources such as dust and indoor air [19].

Dietary habits differ between humans, depending on their socioeconomic status, geographical region, culture, as well as individual choices [15]. Diverse biological responses have been reported as different contaminants are introduced into each human body. Healthy dietary patterns such as the Mediterranean Diet (MedDiet) or the Dietary Approaches to Stop Hypertension (DASH) are evidence-based diets that have been proposed to reduce MASLD prevalence [21,22,23]. According to the European Association for the Study of the Liver guidelines, the macronutrient composition of a diet should include the MedDiet principles and also follow a 500–1000 kcal/day deficit to reduce MASLD rates in the presence of overweight or obesity [24]. These diets are rich in antioxidants found in fruits, vegetables, whole grains, and omega-3 fatty acids mainly found in fish, olive oil, and nuts. Meanwhile, they contain low amounts of sugar, saturated fats, and sodium, which is associated with a lower risk of many chronic diseases. However, food products and groceries, irrespective of their quality, might be tainted with EDCs, mainly resulting from food packaging materials as well as repellents and other chemicals used during cultivation and processing. The Endocrine Society in 2021 emphasised the importance of consuming fresh foods and avoiding processed products as general rules for reducing EDC levels [25]. These recommendations generally align with the basic principles of the previously mentioned dietary patterns.

Recently, lower bisphenol exposure has been linked to greater adherence to the MedDiet among children in Spain and to the Dietary Guidelines for Americans among United States adults [26,27]. However, associations with other EDCs remain undiscovered. Melough et al. used data from the National Health and Nutrition Examination Survey (NHANES) (2013–2016) and examined the correlation between known dietary patterns and EDCs [16]. Interestingly, it was reported that adherence to prudent dietary patterns, such as the MedDiet or the DASH, does not comply with reduced exposure to bisphenols, phthalates, or polycyclic aromatic hydrocarbons. In addition, the exploratory analysis revealed that a 1-point increase in MedDiet adherence was correlated with 2.7% higher (95% confidence interval; CI, 1.7–3.8%) urinary nitrate levels and a 10-point increase in Healthy Eating Index score with 6.8% higher (95% CI, 4.5–9.2%) urinary perchlorate levels and 5.3% higher (95% CI, 2.8–7.9%) nitrate levels accordingly. These findings highlight the unexpected role of an otherwise prudent dietary pattern on EDC exposure as well as the urgent need for monitoring and reducing their concentrations in food supplies and packaging [16].

## 5. EDCs and Developmental Origins of MASLD

### 5.1. Molecular Mechanisms

Hormones bond with receptors in specific target cells and activate them either externally or intracellularly, depending on their water-soluble or lipid-soluble nature [28]. EDCs play a vital role in dysregulating hormonal signaling in the body, acting as mimics or antagonists of them [8]. Certain of them exhibit low accumulation in the human body, such as BPA and phthalates, whereas others, particularly lipophilic ones such as persistent organic pollutants (POPs), tend to accumulate readily in the food chain and adipose tissue [29]. It should be mentioned that EDCs promote disrupting effects even at very low levels of exposure, and they display a non-monotonic response, meaning they can be more harmful in smaller doses. EDCs can interfere with DNA methylation, oxidative stress equilibrium, and hormonal regulation, leading to adverse health effects in humans [30]. According to the developmental origins of health and disease, several EDCs, including BPA, tributyltin (TBT), polychlorinated biphenyls (PCBs), phthalates, perfluorooctanoic acid, and perfluorooctane sulfonate, have been classified as “obesogens” [4]. This derives mainly from experiments on rodents, where neonatal exposure to these chemicals was linked to obesity traits later in life, mainly because they cause augmented fat storage and insulin secretion when introduced into the body. Conversely, phytoestrogens, such as resveratrol, may act protectively against MASLD through engaging with NRs and promoting anti-inflammatory pathways [31]. NRs control a vast amount of the human body’s functions by regulating DNA and ligand binding, related to metabolism and energy/nutrient management as well as organ homeostasis [4,8]. They also exhibit responsiveness to a diverse range of small endogenous ligands, including hormones, vitamins, fatty acids, and metabolites, and they play a vital role in liver metabolism [32]. In humans, 48 NRs have been identified until today [32]. NRs are bonding to elements in the promoter region of the target genes, and then steroid receptor coactivator complexes bind to engage more coactivators with histone-modifying properties that enable gene transcription [4]. When EDCs bind to hormonal ligands, they cause disruption of this signaling system, resulting in changes in the downstream of gene expression, and also interfere with hormonal production or their transportation to target tissues [33]. In this regard, EDCs can lead to the disruption of insulin production in the pancreatic beta cells and the peripheral insulin activity leading to T2DM and IR onset by disrupting the expression of insulin receptor substrate-1 and the activation of the protein kinase B-2 [34]. These mechanisms indirectly lead to MASLD by disrupting the physiological function of the pancreas and, by the subsequent cause of hyperinsulinemia, reduced β-oxidation, and reduced VLDL molecule formation and secretion throughout the body [35].

Although the relationship of EDCs with NRs is well established, their synergistic effect on MASLD needs further examination [33]. Focusing on NRs that directly affect lipid metabolism in the liver (Figure 1), the pregnane X receptor (PXR), the liver X receptor (LXR), and the aryl hydrocarbon receptor (AhR) promote lipogenesis, whereas proliferator-activated receptors (PPARs) and thyroid receptors inhibit hepatic steatosis and promote VLDL secretion [4,32]. Also, steroid receptors such as the oestrogen receptor (ER) and the androgen receptor (AR) decrease lipid accumulation, whereas the glucocorticoid receptor promotes liver damage and steatosis. Most of these NRs act through genomic signaling pathways by directly engaging promoter regions in the target gene, whereas several, such as ERα, ERβ, the AR, and the progesterone receptor (PR), also have roles in non-genomic signaling. This means that their subsequent activation by an EDC in the cytoplasm results in mitogen-activated protein kinase (MAPK) and/or phosphoinositide 3-kinase (PI3K)/protein kinase B (PKB or AKT) pathways activation, transcription factor engagement, and altered gene transcription in the downstream [4,34].

As of the latest available knowledge, perfluorinated compounds have the capability to interact with the ER and PPARα, act antagonistically to their action, and produce a downstream pathway resulting in dysfunction and steatosis in liver cells. Specifically, when the PPARα is activated by its natural ligands (free fatty acids), it leads to the expression of the carnitine palmitoyltransferase 1 and acyl-CoA oxidase 1 genes that contribute to lipid oxidation, whereas when activated by EDCs, it increases the expression of the PXR gene, which leads to hepatic lipid accumulation [4]. Furthermore, some PAHs and/or dioxins can activate molecular pathways towards hepatic steatosis by binding to the AhR in the liver. In addition, BPA acts through linking NRs such as ERs, AR, and/or membrane receptors such as the G protein-coupled estrogen receptor (GPER), resulting in activation of MAPK and PI3K/AKT pathways, promoting the transcription of mRNA of proinflammatory mediators such as interleukin-1 (IL-1), IL-6, and tumor necrosis factor a (TNFa), all reported as basic predecessors of inflammation linked to metabolic syndrome [34]. On the contrary, in a study including a hepatic cell line lacking ERα expression and with minimal ERβ expression, BPA was found to enhance lipid accumulation and reduce lipid secretion, potentially via negative regulation of the PI3K/Akt signaling pathway [35]. In fact, BPA caused a significant reduction in mRNA expression of the PPARγ and PPARβ/δ genes, which resulted in reduced regulation of lipid homeostasis and reduced mRNA expression of the apolipoprotein B, a vital protein for VLDL secretion from the liver to the blood circulation.

BPA is a rapidly metabolised chemical in the adult body, though its metabolic rate is extremely lower in infants, suggesting that early-life exposure, even at lower concentrations, is far more dangerous than during adulthood [36]. As previously mentioned, BPA is found to either upregulate or downregulate the PI3K/AKT pathway, both of which result in inflammation, disrupted β-oxidation, and lipid accumulation in the liver [37]. On the contrary, phytoestrogens, such as resveratrol, can act protectively against MASLD by also downregulating the same pathway. In a study by He and his colleagues, treatments with curcumin and resveratrol in rats indicated a reduction in the signaling of the PI3K/AKT/mTOR pathway, followed by a downregulation of the mRNA of hypoxia-inducible factor 1-alpha and vascular endothelial growth factor (VEGF), both known to promote angiogenesis. Notably, the synergistic use of these antioxidants was found to improve the texture of livers and reduce lipid droplets in the treatment group (that received treatment with either curcumin, resveratrol, or both) compared to the control group, potentially contributing to MASLD prevention [37]. Another study revealed their synergistic cytoprotective impact against oxidative stress by diminishing ROS levels and fostering antioxidant regulators such as the nuclear factor erythroid factor 2-related factor 2, heme oxygenase 1, superoxide dismutase, and nicotinamide-adenine dinucleotide (NAD) [31].

### 5.2. Impact on Mitochondrial Function

Each human cell contains thousands of mitochondria, which are responsible for energy production from the breakdown of glucose or fatty acids [38]. It has been recently proposed that mitochondrial changes in terms of genes, energetics, and morphology caused by EDCs may also play an important role in MASLD onset [33]. Specifically, oxidative stress-related pathways induced by EDCs could produce mitochondrial disruption, resulting in ROS disequilibrium, lipid accumulation, and inflammation, highly present metabolic dysfunctions in MASLD [7]. Furthermore, mitochondrial DNA repair mechanisms are poorer compared to nuclear DNA, making it more susceptible to damage when interacting with EDCs [7,39]. For instance, EDCs are prooxidant substances that can alter ROS homeostasis and act through ROS-mediated pathways towards mitochondrial endpoint disruption. Another possible way of action is through causing damage to mitochondria or their biogenesis, resulting in reduced cell capacity to deal with excessive ROS amounts. Additionally, EDCs can disrupt mitochondrial stimulation through interfering with hormonal actions and changing downstream signaling pathways [40]. Thus, mitochondrial functionality tests might be a useful tool for the recognition of specific EDCs in the future.

Ying et al. (2014) performed a longitudinal examination of mitochondrial function in rats exposed to BPA during the perinatal period. Their study demonstrated a reduction in mitochondrial respiratory complex activity starting at three weeks of age, which continued progressively, resulting in hepatic oxidative stress, accompanied by an increase in steatosis [30]. This exposure to low BPA levels (lower than 40 g/kg/day, which is within the tolerable daily intake) led to the development of steatosis in rats after birth. In another study, Khan et al. evaluated hepatic mitochondria function in rats and observed decreased activities of enzymes and the complexes I–V in response to BPA exposure, mainly due to disrupted ROS-mediated pathways [41]. Furthermore, it has been reported that EDCs target the steroidogenic acute regulatory (StAR) protein, which normally serves as a steroid synthesis regulator by managing the cholesterol influx from the outer to the inner mitochondrial membrane. For instance, Pu et al. found that when ovarian theca cells were exposed to TBT, they upregulated the StAR mRNA, resulting in cholesterol efflux, leading to disrupted mitochondrial functionality [42,43]. In conclusion, EDCs promote oxidative stress disequilibrium and interfere with mitochondrial DNA, thus resulting in mitochondrial endpoint disruption and disrupted functionality, altogether contributing to MASLD onset [33,40].

### 5.3. Epigenetic Regulations and Transgenerational Inheritance

EDCs seem to interfere with and alter epigenetic mechanisms leading to disease inheritance, and the main mechanisms are presented in the figure below [28] (Figure 2). Nowadays, DNA methylation is the most frequently studied epigenetic regulation resulting from EDCs [38]. It has been proposed that DNA methylation could stratify medium-to-severe MASLD fibrosis and be used as a valuable non-invasive tool in the future [44,45]. The main proteins that participate in DNA and histone methylation are the histone methyltransferase and the DNA methyltransferase (DNMT) [46]. Hepatic DNMT levels have been found to be increased in individuals with steatohepatitis, whereas lower levels have been noticed in those with simple steatosis. Over the past few years, many genes have been identified to undergo altered transcription following BPA exposure and subsequent changes in DNA methylation. Specifically, Kundakovic et al. found that following in utero BPA exposure, the ERa, and the brain-derived neurotrophic factor genes, related to nervous system functionality, were hypomethylated in the brains of rat descendants, promoting neurodevelopment [47,48]. In another experimental study, the signal transducer and activator of transcription 3 (STAT3), which plays a fundamental role in immune signaling, was found to be hypermethylated after BPA exposure [47]. In addition, it has been reported that chronic BPA exposure is found to increase hepatic cholesterol and triglyceride levels through promoting hypomethylation and thus increasing the expression of the transcription factors sterol regulatory element binding transcription factor 1 (SREBF1) and SREBF2, which upregulate genes related to lipid synthesis [46,49]. The impact of DNA hypomethylation on lipid accumulation in the liver following BPA exposure underscores their synergistic effect in MASLD development.

Histone modifications refer to acetylations/deacetylations catalysed by histone acetyltransferases (HAT) and histone deacetylases (HDAC) [50]. From experiments on rodents, it seems that the imbalance between HAT and HDAC in the liver is associated with liver damage, as well as that histone modifications contribute to IR and therefore MASLD [51]. EDCs are known to interfere with NRs by altering their structure to be more compatible with coactivators. Some coactivators possess HAT activity, meaning they can both increase the transcription of other coactivators and other genes [51]. Enhanced HAT activity is linked to abnormal gene overexpression, potentially contributing to unusual development and disease [52]. In their study, Azumi et al. reported that when ascidian *Ciona intestinalis* were treated with TBT, an unprecedented expression of 130 genes was noticed, and Nakanishi et al. found that choriocarcinoma cells increased mRNA levels of the aromatase cytochrome P450 gene, a key enzyme in the biosynthesis of oestrogens, due to the same exposure [52,53,54]. These alterations were mainly attributed to increased HAT activation induced by TBT [51,54]. Furthermore, it has been reported that BPA and DES exposure can increase histone H3K4-trimethylation, the zeste homolog 2, a specific histone methyltransferase, and RNA polymerase II in MCF7 cells, as well as ERs and ER-coregulators (histone methylases as MLL1 and histone acetylases as p300), all resulting in increased gene transcription [47]. Shifting the focus to HDACs, NAD-dependent deacetylase sirtuin-1 is a major hepatic metabolism modulator acting by deactivating the peroxisome proliferator-activated receptor-gamma coactivator 1 gene, known to promote gluconeogenesis and inhibit glycolysis, and the PXR gene, which is known to promote steatosis in the liver [55]. Some EDCs, such as methoxyethanol, act as DNMT inhibitors, resulting in DNA methylation and unsupervised histone acetylations that progressively lead to MASLD through the aforementioned pathways [23,55].

Many researchers support that early life exposure to EDCs could be a crucial risk factor for MASLD onset and progression later in life [16,33]. In fact, children who experienced the Great Chinese Famine during the gestational period had significantly greater odds (odds ratio = 1.10, 95% CI 1.00–1.21) for developing MASLD in adulthood compared to the nonexposed participants [56]. Also, newborn babies from mothers with IR had increased odds of developing hepatic steatosis, supporting the hypothesis of the effect of the intrauterine environment on MASLD onset [4,57]. Early-life exposure to EDCs, even in small amounts, can be penetrative through generations. Studies have shown that neonatal and intrauterine BPA exposure has been linked to the presence of altered methylation in CpG sequences, histone modifications, and differences in mRNA levels [47,58]. In addition, experiments on rodents have revealed that tributyltin exposure could induce the transgenerational inheritance of obesity and hepatic steatosis [59]. Chamorro Garcia et al. discovered that pregnant rats’ exposure to the obesogen TBT caused epigenetic alterations that were transgenerationally inherited until the F3 descendants. This refers to the augmented size and number of the white adipose tissue depot as well as the reprogrammed mesenchymal stromal stem cells towards becoming adipocytes, which appear in the F1 and the rest generations, both contributing to MASLD formation [60]. Precisely, TBT mainly increased the mRNA levels of PPARγ (master regulator of adipogenesis in mesenchymal stromal stem cells), fatty acid binding protein 4, and zinc finger protein 423 (both activating PPARγ). Evidence from animal models has shown that transgenerational inheritance of diseases has been observed due to EDC exposures through sperm epimutations [38]. Precisely, only the F0 female rats had been exposed to the chemical. Afterwards, when the control and the methoxychlor-exposed lineage F3 rats were compared, 37 significant epimutations were identified in the sperm. Further examination of these epimutations in relation to other EDC exposures showed that only a few overlapped with the ones found after methoxychlor exposure. This indicates that the transgenerational epimutations are specific to methoxychlor exposure. In another study from Ma et al., BPA exposure during the gestational period resulted in increased body weight, IR, and glucose intolerance in the descendants [46,61]. This was probably caused by BPA-promoted hepatic DNA hypomethylation together with glucokinase gene hypermethylation, a protein that regulates glycolysis in the liver. The researchers proposed that the suppression of glucokinase function is the underlying mechanism leading to the development of IR and T2DM, key “hits” in the pathogenesis of MASLD.

Regarding another crucial epigenetic regulator, miRNA molecules also contribute significantly to the progression of MASLD and are subject to dysregulation by EDCs [62,63]. Multiple miRNAs have been found to trigger MASLD through promoting lipid accumulation, trafficking, catabolism, ROS generation, and carbohydrate metabolism disruption in hepatic cells [44,63]. In fact, they have been suggested to serve as non-invasive biomarkers that stratify disease severity. miRNAs participate in hepatic cell metabolism, including glycogen metabolism (miRNA-122, miRNA-29, miRNA-20), lipogenesis (miRNA-122, miRNA-33), β-oxidation (miRNA-122, miRNA-33, miRNA-21), and others [62]. The latest available scientific data related to MASLD support that miRNA-122, miRNA-33, miRNA-34a, and miRNA-21 are the most prominent molecules that contribute to disease development and progression. For instance, miRNA-21, mainly elevated in MASLD patients, downregulates the phosphatase and tensin homolog, which inhibit de novo lipogenesis and fatty acid uptake by the liver, as well as PPARα, which normally promotes lipid oxidation. On the contrary, miR-122, which targets the glycolytic enzyme aldolase A, and miR-34a, which targets lactate dehydrogenase A, are both downregulated in patients with MASLD, resulting in inhibited glycolysis and glucose accumulation in the liver. miRNA-33, which targets phosphoenolpyruvate carboxykinase and glucose-6-phosphatase, inhibits gluconeogenesis, which is a common clinical characteristic of MASLD patients [62].

It has been reported that EDCs can regulate the expression of important miRNAs for human metabolism. For instance, De Felice et al. found that miRNA-146a was overexpressed in the placenta of pregnant women living in polluted areas and highly exposed to BPA, compared to women living in healthy environments [64]. miRNA-146a targets genes involved in metabolic pathways, the MAPK pathway, and endocrine system genes as the mediator complex subunit 1, which interacts with AR, ERa, and p53, all of which are involved in liver metabolism. Also, 2,3,7,8 tetrachlorodibenzo-p-dioxin, a widely toxic chemical, regulates miRNA-122 expression and miRNA-101a, which downregulates cyclooxygenase-2, a factor typically associated with liver damage [28]. Interestingly, Craig et al. found that when female zebrafish were exposed to fluoxetine, they presented augmented hepatic expression of the miRNA-22b, miRNA-140, miRNA-210a, and miRNA-301, which normally silence pathways related to insulin signaling, de novo lipogenesis, as well as cholesterol and triglyceride synthesis [28,65]. Lastly, in another study, DDT-exposed rats presented decreased hepatic levels of miRNA-21, 221, 222, and 429 expressions, which are possible regulators of the CYP proteins CYP1A and CYP2B. CYP are ligands for specific receptors in the liver, such as Ahr and PXR, and promote gene expression, as previously mentioned, related to steatogenesis [28]. Decreased levels of these miRNAs are correlated with increased mRNA levels of their target genes, CYPs, thus unprecedented pathways related to steatogenesis.

### 5.4. EDCs and Their Role on MASLD Progression

In MASLD, lipid accumulation and reduced β-oxidation are mainly observed in the liver, referred to as a reversible state, whereas their progression leads to metabolic dysfunction-associated steatohepatitis (MASH) and hepatocellular carcinoma (HCC), which are basically characterised by irreversible fibrosis [4,33]. Progressive dysfunction in mitochondrial energetics, ROS disequilibrium, macrophage infiltration, and inflammation, as well as reduced insulin signaling and IR, are common characteristics in these MASLD progression states [4].

It is reported that EDC exposure can lead to MASLD progression, MASH, and hepatocellular carcinoma [33]. Many studies have reported the carcinogenic nature of another major class of EDCs, heavy metals. Precisely, in a study including rat liver cells, when exposed to sodium arsenite, augmented DNA hypomethylation related to hepatic malignancies was observed [33,66]. This was also reported in another study on rodents, where increased steatogenesis, overexpression of genes related to carcinogenesis, as well as global liver DNA hypomethylation were noticed after the exposure [67]. As previously mentioned, BPA exposure can mediate STAT3 gene expression, inducing dose dependent changes in its methylation [47]. Interestingly, in a study including rats exposed to BPA, changes in STAT3 gene methylation occurred prior to the onset of tumors, implying that this gene may be an early indicator of hepatic tumorigenesis [68]. Additionally, Sun L. et al. studied acute and chronic zebrafish exposure to triclosan and BPA, and they discovered that they disrupted lipid metabolism and contributed to MASLD progression [69]. Precisely, BPA exposure resulted in upregulated expression of genes involved in lipogenesis, such as the fatty acid synthase (FASN), the enhancer-binding protein-alpha (CEBPA), the SREBF1, and the acetyl-CoA carboxylase 1 (ACC), whereas downregulated genes involved in β-oxidation, such as the acyl-CoA oxidase 1, the carnitine palmitoyltransferase 1A, and the PPARa genes. As for the long-term exposure to triclosan, expression of inflammation-related genes such as IL-1 and TNFa was significantly upregulated, whereas long-term exposure to BPA induced overexpression of IL-1b, nuclear factor kappa-light-chain-enhancer of activated B cells (involved in inflammation), C/EBP homologous protein, protein kinase R-like endoplasmic reticulum kinase, activating transcription factor 6, binding immunoglobin protein, and iron responsive element [causing endoplasmic reticulum stress] genes. Taken together, exposure to these chemicals can not only induce MASLD but also contribute to MASLD progression through inflammation, promoting lipogenesis, reducing β-oxidation, and causing endoplasmic reticulum stress [69].

Figueiredo et al. were the first to show that when ovariectomised mice were exposed to BPA, they presented increased fat accumulation and collagen deposition in the liver [70]. Specifically, SREBF1 and stearoyl-CoA desaturase genes were significantly upregulated in the livers of BPA-exposed overectomised mice when compared to control ovariectomised mice. Also, BPA exposure induced ER stress in the liver of the female rodents through increasing the hepatic expression of related genes such as heat shock protein family A (Hsp70) member 5, hypoxia up-regulated protein 1, and activating transcription factor 6 compared to controls (*p* < 0.05). Altogether, this study may indicate that exposure of this chemical to women, especially during physiological loss of ovarian function later in life, is linked to MASLD progression by the subsequent dysregulation of homeostasis and induced stress in the liver [70]. Additionally, Massart and colleagues reported in their review that environmental contaminants are suspected to worsen MASLD and contribute significantly to its progression, but the direct mechanisms need to be elucidated [71]. In addition, BPA exposure causes SREBP1 overexpression in hepatic cells, leading to triglyceride accumulation [72], as well as increasing SREBP2 and causing cholesterol accumulation in the liver [73]. Also, perfluorooctanoic acid exposure may be linked to increased ALT levels, augmentation of triglycerides, and steatosis, although further studies are needed to confirm this statement [71].

### 5.5. Gene–EDCs Interactions in MASLD

The science of gene–environment interactions examines the crosstalk between the two distinct fields, aiming at defining an individual’s susceptibility to adverse health effects when being exposed to different environmental agents [38]. To examine these interactions, data from consortia or individual studies, relying on large sample sizes for genome-wide association studies, are used. It has been reported that the interplay between the genome, the epigenome, and environmental factors could contribute to MASLD susceptibility and progression [74]. It is well established that not every patient with MASLD develops steatosis, chronic inflammation, or liver-related morbidity. Previous work from our team has revealed that gene–diet interactions play a vital role in MASLD susceptibility [75,76]. However, there is still a paucity of evidence from studies regarding gene–EDC interactions in MASLD.

Existing evidence suggests that gene–EDC interactions lead to diverse health effects that vary based on individuals’ characteristics and may be linked to metabolic diseases [77]. Unravelling one’s susceptibility to EDCs and the resulting changes could lead to new regulatory mechanisms, create new preventive measures, and also reduce the negative effects of exposure. Dunaway and his colleagues in 2016 examined the potential connection of autism spectrum disorder (ASD) with the interaction between the genome and exposure to PCB [78]. This interaction was positively correlated with ASD due to altered PCB-related methylation of the DNA. In another case, a genome-wide association study was performed in a Chinese population to examine possible correlations between the genetic variants of some oxidative stress markers and three urinary EDCs, including BPA, BPF, and triclosan [79]. Interestingly, gene–EDC interactions were found to modify individuals’ responses to oxidative stress based on genetic modifications caused by these chemicals. Although gene–EDC interactions have been reported in ASD, oxidative stress, and fertility [80], their impact on MASLD still remains an undiscovered field and needs further examination.

## 6. Practical Recommendations for EDCs Exposure Reduction & MASLD Prevention

The prevention of MASLD can be accomplished through various means, given that its origins are influenced by environmental factors, lifestyle choices, genetic predisposition and transgenerationally inherited alterations [4]. Whereas food is infiltrated with EDCs, without being referred to in food labels, there are some basic practices that can be followed to reduce the exposure [20,81]. Focusing on factors within human control, the following table (Table 1) includes fundamental recommendations provided by the National Institute of Health and from other recent updated literature sources that can contribute to diminishing EDCs levels through dietary and everyday practices [82,83]. Given that these contaminants contribute to MASLD onset, these recommendations could serve as a proactive step towards lowering its prevalence worldwide. It is important to highlight that this guide should be embraced by all individuals, regardless of their susceptibility to MASLD, with special attention to pregnant women, infants, and children.

## 7. Concluding Remarks

In recent years, MASLD has gained much attention as the primary contributor to chronic liver disease and inflammation, as well as liver-related morbidity and mortality. The current review introduced a fresh outlook on MASLD by delving into the interplay between EDCs, epigenetic alterations, and interactive molecular mechanisms. Although the combination of weight loss with physical activity has been proposed as the gold standard to reduce liver steatosis [21,22,23], the scientific community must proactively investigate other potential contributors. As apparent from the most up-to-date literature, greater exposure to EDCs, even from the in utero environment, is highly correlated with an increased probability of developing MASLD in adult life [4,33,84].

Concerning dietary practices, MedDiet has been proposed as the most suitable dietary regimen for MASLD, although more studies are needed to confirm this statement [85]. This diet’s basic principles are in absolute agreement with the ones aimed at reducing the impact of EDCs. Favoring the consumption of fresh products and avoiding red meat rich in fat and canned products, burnt parts, and ready-to-eat meals are major practices for reducing the impact of EDCs, favoring MedDiet principles, and reducing MASLD rates.

Focusing on unravelling the pathophysiological mechanisms leading to MASLD onset is crucial, even in the gestational period [14,58,84]. EDC-induced epigenetic alterations from this early life stage are heightening the probability of increased MASLD susceptibility across the next generation. As previously mentioned, numerous EDCs have been characterised as obesogens, capable of triggering MASLD onset [14,60]. Current available knowledge suggests that the most prominent linking pathways are the induced oxidative stress disequilibrium resulting in mitochondrial changes and the interaction with NRs resulting in modified gene transcription [4]. They can also cause non-genomic signaling through activating the PI3K/AKT or MAPK/ERK pathways, resulting in the expression of target genes related to inflammatory pathways, IR, and lipid accumulation [4,34,37]. Regarding epigenetic alterations, EDCs can regulate gene expression of proteins involved in DNA methylation and histone modifications [38,44,45]. In addition, EDC exposure is also linked to miRNA regulation, leading to dysfunction of related pathways, causing liver damage, and disrupting liver homeostasis [28]. Probably, miRNAs could be used as EDC-specific exposure biomarkers for MASLD in the future.

In conclusion, establishing non-invasive MASLD treatments nowadays remains a challenge. Further research with large cohorts of patients, including data on genetic profiles, the epigenome, biochemical markers, and lifestyle traits of risk individuals, is a crucial step in preventing MASLD, as is promoting new holistic approaches towards precise medicine; these will contribute significantly to the reduction of MASLD and associated diseases’ rates [1,2].

## Figures and Tables

**Figure 1 nutrients-16-01124-f001:**
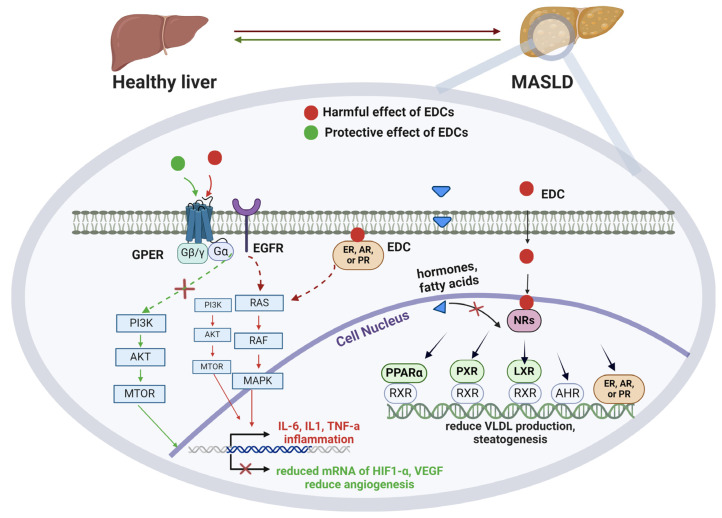
Genomic and non-genomic signaling of EDCs towards MASLD. The genomic signaling takes place in the nucleus, where EDCs bind to NRs as such PPAR, PXR, LXR that bind to RXR or directly to AHR, ER, AR, and PR and promote gene transcription. This way, EDCs are antagonising physiological ligands, fatty acids, or hormones, resulting in the enhancement or inhibition of hepatic steatosis. The non-genomic signaling mainly involves GPERs that are activated by EDCs, activating EGFR, and promoting MAPK or the PI3K/AKT pathways. This leads to induction of gene transcription, leading to production of inflammation-related cytokines, such as IL-6 and TNFa, promoting IR and associated diseases. In addition, ER, AR, or PR receptors can be activated in the cytoplasm by EDCs and promote the aforementioned pathways. On the contrary, some EDCs such as phytoestrogens have cytoprotective effects, and they can inhibit the PI3K/AKT pathway, resulting in reduced transcription of the mRNA of HIF1-α and VEGF genes, both known to promote angiogenesis. EDCs; endocrine disrupting chemicals, NRs; nuclear receptors, PPAR; proliferator-activated receptors, PXR; pregnane X receptor, LXR; liver X receptor, RXR; retinoid X receptor, AHR; aryl hydrocarbon receptor, ER; oestrogen receptor, AR; androgen receptor, PR; progesterone receptor, GPER; G protein-coupled estrogen receptor, EGFR; epidermal growth factor receptor, MAPK; mitogen-activated protein kinase, PI3K/AKT; phosphoinositide 3-kinase, IL-6; interleukin 6, TNFa; tumor necrosis factor a, HIF1-α; hypoxia-inducible factor 1-alpha, VEGF; vascular endothelial growth factor. Created with BioRender.com, accessed on 5 April 2024.

**Figure 2 nutrients-16-01124-f002:**
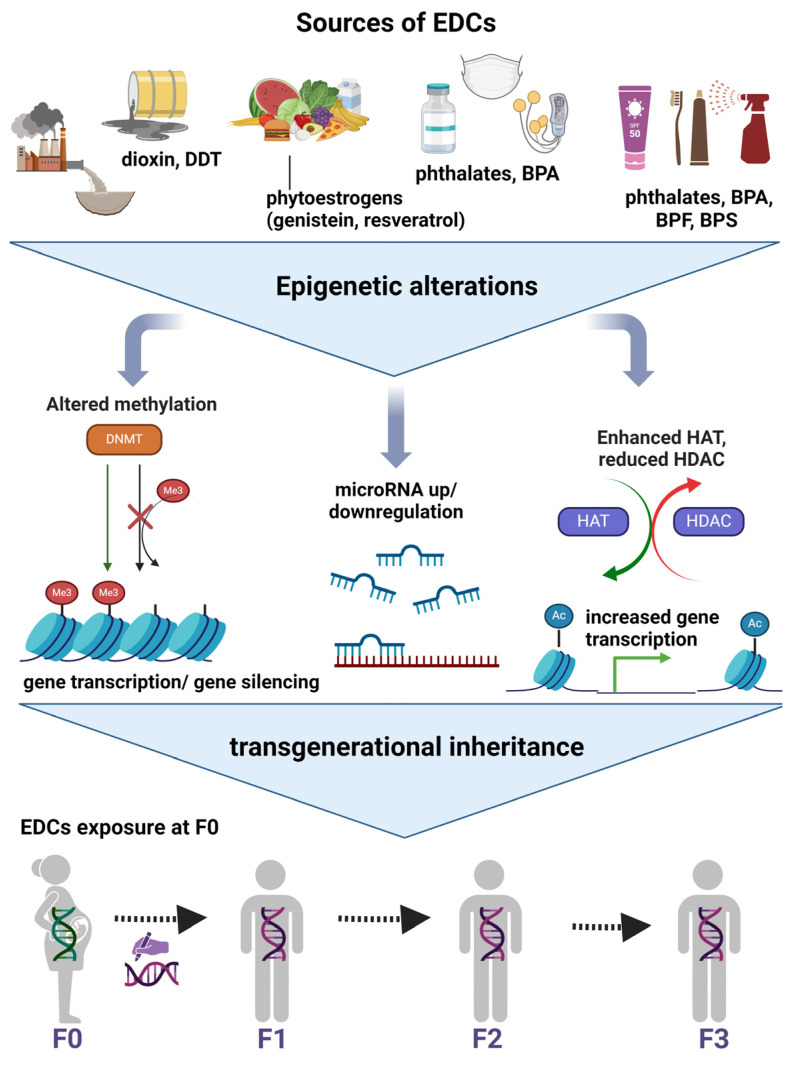
Transgenerational inheritance. In this figure, the most prominent sources of EDCs are presented. Exposure to these chemicals during the gestational period, childhood, or later in life is linked to produced epigenetic alterations, such as altered methylation, microRNA regulation, and histone acetylation. The aforementioned epigenetic alterations can be really penetrative across generations. DDT; dichlorodiphenyltrichloroethane, BPA; bisphenol A, BPF; bisphenol F, BPS; bisphenol S, DNMT; DNA methyltransferase, HAT; histone acetyltransferases, HDAC; histone deacetylases. Created with BioRender.com, accessed on 5 April 2024.

**Table 1 nutrients-16-01124-t001:** Recommendations on reducing EDC exposure through diet and everyday practices for adults and babies.

**Dietary recommendations**
Prefer fresh fruits, vegetables, fish and preferably seasonal products.
Buy tomato sauce stored in glass jars and not plastic bottles.
Avoid using popcorn bags in the microwave, instead cook the popcorn in the pot.
Choose products that are not stored in plastic containers or cans, as ready-to-eat and fast-food products.
Prefer sandwiches or bakery products unwrapped, just freshly exposed at the countertop.
Avoid burned, smoked or pre-fried foods and, when possible, remove the burned parts (such as meat, meat by-products and canned fish).
Avoid frozen fish, pizza, and other foods.
Avoid packed meals such as instant soups and noodles.
Avoid the use of plastic coffee or tea bags, when not certified as EDC free and use loose tea/coffee.
**Cooking, kitchen articles & food storage**
Avoid using non-stick cooking utensils if the coating is worn.
Avoid grilling, barbecuing, deep frying and overheating of food.
Consult the manufacturer’s instructions when packaging food and use grease-proof paper or film.
Use plastic food containers only in perfect condition.
Always follow the manufacturer’s instructions and use only plastic bottles or cans in perfect condition, when heating food and/or milk for babies.
When using plastic containers, not labelled as appropriate for high temperatures, only pour cold food or beverages.
Avoid using polycarbonate baby bottles for milk/water.
Don’t put plastic containers to be heated in the microwave/oven.
**Products, proper ventilation and house practices**
Avoid buying materials with soft PVC that contain di (2-ethylhexyl) phthalate or DEHP.
Avoid the use of soft PVC floors for children. Instead use a carpet made of untreated fibres (i.e., wool, cotton).
Avoid PVC objects/toys for children.
Substitute worn out or damaged wrappings of objects having foam padding such as car seats and mattresses.
Avoid candle smoke or cigarette smoke in your living environment.
Make sure of proper house ventilation to avoid dust accumulation indoors and properly maintain your vacuum cleaner (change filters, empty the dust bag, etc.).
Buy clothing products with known origins and composition instead of waterproof or anti-stain labels.
Check product labels and avoid beauty products with parabens and/or phthalates.
Use water-based sunscreen to avoid the use of chemicals.
Avoid fragrances, body sprays, perfumes and the unnecessary use of cosmetics, cleaning, dental care products.
Avoid using scented or antibacterial hand soaps, candles, or air fresheners.

References [20,81,82,83].

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
