# Peer review of "The Interplay between Endocrine-Disrupting Chemicals and the Epigenome towards Metabolic Dysfunction-Associated Steatotic Liver Disease: A Comprehensive Review"

_nutrients, 2024, doi:10.3390/nu16081124_

Round 1

Reviewer 1 Report

Comments and Suggestions for Authors

The title of this article is “The interplay between endocrine-disrupting chemicals and the pigenome towards Metabolic Dysfunction-Associated Steatotic Liver Disease: a comprehensive review”. This is an interesting topic. However, there are still some areas of the article that need to be revised:

1.      In the Introduction section, the phrase "To our knowledge" in line 49 is inappropriate, and it is suggested that it be changed to "Herein".

2.      Please revise the title of the third part of the article. It is not recommended to use "result" in the analysis and discussion of the review.

3.      Please use abbreviations, e.g. WHO, when they appear more than three times in the article.

4.      In section 3.3, should the molecular mechanism be introduced before 3.2.2 " Practical recommendations for EDCs exposure reduction ".

5.      In part 3.3.2 should not end with an example, but the author should summarize the effect of EDC on mitochondrial function at the end of the paragraph.

6.      Part 3.3.5 is about gene-environment interactions in MASLD, but only the first paragraph mentions one sentence about the environment, and the second paragraph is about gene-EDC interactions, so the authors should check whether the title is consistent with the content.

7.      In the conclusion part, "Reducing the disease burden worldwide." is too general a statement, so please describe the key role of MASLD prevention.

Comments on the Quality of English Language

 Minor editing of English language required

Author Response

  1. Comment: In the Introduction section, the phrase "To our knowledge" in line 49 is inappropriate, and it is suggested that it be changed to "Herein".

Answer: Thank you for this comment. As suggested, we replaced the phrase "To our knowledge" in line 54 in the introduction to "Herein".

  1. Comment: Please revise the title of the third part of the article. It is not recommended to use "result" in the analysis and discussion of the review.

Answer: As very rightfully proposed, we revised the sections of the third part of the article. Specifically, we erased the title ‘’results’’ and changed the title ‘’discussion’’ to ‘’concluding remarks’’. Afterwards, the sections were revised and different numbers were put in each section accordingly.

  1. Comment: Please use abbreviations, e.g. WHO, when they appear more than three times in the article.

Answer: Thank you for this meaningful comment. We updated all the abbreviations in the texts based on their appearance (Didn’t use an abbreviation if it didn’t appear more than three times in the text and erased some of them in the ‘’Abbreviations’’ section).

  1. Comment: In section 3.3, should the molecular mechanism be introduced before 3.2.2 " Practical recommendations for EDCs exposure reduction ".

Answer: Totally agree with this point of view. We changed the order of the paragraphs upon this comment and the paragraph ‘’6. Practical recommendations for EDCs exposure reduction & MASLD prevention’’ (line 939) was moved after the section ‘’5. EDCs and developmental origins of MASLD’’ (line 193).

  1. Comment: In part 3.3.2 should not end with an example, but the author should summarize the effect of EDC on mitochondrial function at the end of the paragraph.

Answer: Thank you for pointing this out. It was totally necessary for this section to include a concluding sentence at the end of the section ‘’5.2 Impact on mitochondrial function’’ (lines 597-600): ‘’In conclusion, EDCs promote oxidative stress disequilibrium and interfere with mitochondrial DNA, thus resulting in mitochondrial endpoints disruption and disrupted functionality, altogether contributing to MASLD onset [33,40]’’.

  1. Comment: Part 3.3.5 is about gene-environment interactions in MASLD, but only the first paragraph mentions one sentence about the environment, and the second paragraph is about gene-EDC interactions, so the authors should check whether the title is consistent with the content.

Answer: After taking into consideration this comment, the title of the subsection 5.5 was shifted from ‘’gene-environment interactions in MASLD’’ to ‘’gene-EDCs interactions in MASLD’’.

  1. Comment: In the conclusion part, "Reducing the disease burden worldwide." is too general a statement, so please describe the key role of MASLD prevention.

Answer: Thank you for your contribution to this manuscript and for pointing out to revise some critical points. In the conclusion, the phrase "Reducing the disease burden worldwide" is substituted by ‘’ aimed at reducing MASLD and associated diseases’ rates worldwide that are currently showing an upward trend’’.

Reviewer 2 Report

Comments and Suggestions for Authors

The manuscript presents a competently written synthetic review of an important subject. Endocrine-disrupting chemicals are important pollutants, present not only in food but also in food products, where they are produced by food processing but also penetrate from packages and vessels. All these aspects are discussed in the text.

Materials and Methods are presented, not in a full detail (including the number of articles found and considered etc) but generally sufficiently.

Conclusions are well formulated and reasonable

List of abbreviations is useful as abbreviations, once introduced, often appear in distant regions of the text.

Remarks:

Lines 253/254: “found improve”, perhaps “found to improve”?

Lines 287/288: “mitochondrial DNA lacks […] repair mechanism”, this view, although often repeated, is not true; indeed, mitochondrial repair is poorer (with respect to nuclear DNA) but not absent (e.g. Muftuoglu, M., Mori, M. P., & de Souza-Pinto, N. C. (2014). Formation and repair of oxidative damage in the mitochondrial DNA. Mitochondrion, 17, 164-181; Gredilla, R., Bohr, V. A., & Stevnsner, T. (2010). Mitochondrial DNA repair and association with aging–an update. Experimental gerontology, 45(7-8), 478-488).

Line 342: “Ciona intestinalis”, please in italics

Line 390: “overlapped with methoxychlor”, perhaps not clear enough

Figures 1 and 2 are blurry (low resolution) but it may be an artefact of pdf formation. Lower part of the Figure: is the DNA damage symbolized clearly enough?

Lines 448/449: “where they presented”, who “they”?

Line 470: “Figueiredo L.S. et al “, are the initials necessary?

Line 488: “Gene x environment”?

Author Response

    1. Comment: Lines 253/254: “found improve”, perhaps “found to improve”?

    Answer: Thank you for indicating this. It was missed out. In Line 513 of the manuscript: the phrase “found improve” is now changed to “found to improve”.

    1. Comment: Lines 287/288: “mitochondrial DNA lacks […] repair mechanism”, this view, although often repeated, is not true; indeed, mitochondrial repair is poorer (with respect to nuclear DNA) but not absent (e.g. Muftuoglu, M., Mori, M. P., & de Souza-Pinto, N. C. (2014). Formation and repair of oxidative damage in the mitochondrial DNA. Mitochondrion, 17, 164-181; Gredilla, R., Bohr, V. A., & Stevnsner, T. (2010). Mitochondrial DNA repair and association with aging–an update. Experimental gerontology, 45(7-8), 478-488).

    Answer: After taking into consideration the proposed literature and your valuable input in this paragraph we changed this sentence according to the most prominent view about mitochondrial DNA; specifically, from ‘’lacks’’ to ‘’is poorer’’ concerning its repair mechanisms (Lines 566-568): ‘’Furthermore, mitochondrial DNA repair mechanisms are poorer compared to nuclear DNA, making it more susceptible to damage when interacting with EDCs’’. Additionally added a new literature (42) in line 568.

    1. Comment: Line 342: “Ciona intestinalis”, please in italics

    Answer: As proposed, in line 649 “Ciona intestinalis” was changed to “Ciona intestinalis” (italics)

    1. Comment: Line 390: “overlapped with methoxychlor”, perhaps not clear enough

    Answer: Thank you for making this comment. The phrase was replaced in order to be clear enough and avoid possible confusions. Specifically, in lines 707-708: the phrase “overlapped with methoxychlor” was changed to ‘’only a few overlapped with the ones found after methoxychlor exposure’’.

    1. Comment: Figures 1 and 2 are blurry (low resolution) but it may be an artefact of pdf formation. Lower part of the Figure: is the DNA damage symbolized clearly enough?

    Answer: Thank you for pointing out some changes concerning the figures, it was totally necessary to indicate somehow the transgenerational inheritance across generations that was missing before. Specifically, at the end of the figure 2 according to your valuable comment, we tried to point out that mother’s DNA after been imprinted with epigenetic alterations after EDCs’ exposure (green DNA before), is transgenerationally inherited for 3 generations (changed color; purple after exposure). Additionally, the figures were previously presented in png files with 300 dpi but in any case, we replaced them with jpeg files also having 300 dpi (high resolution).

    1. Comment: Lines 448/449: “where they presented”, who “they”?

    Answer: Thank you for pointing out the missing information about this sentence. According to this comment, in lines 811-813 the sentence “This was also reported in another study on rodents, where they presented increased steatogenesis after the exposure, overexpression of genes related to carcinogenesis as well as global liver DNA hypomethylation” is changed to ‘’ This was also reported in another study on rodents, where increased steatogenesis, overexpression of genes related to carcinogenesis as well as global liver DNA hypomethylation were noticed after the exposure’’.

    1. Comment: Line 470: “Figueiredo L.S. et al “, are the initials necessary?

    Answer: In line 894: “Figueiredo L.S. et al “is changed to “Figueiredo et al’’ as you appropriately pointed out.

    1. Comment: Line 488: “Gene x environment”?

    Answer: In line 913: “Gene x environment” is changed to ‘’Gene-environment’’.